# Intragenomic variation in non-adaptive nucleotide biases causes underestimation of selection on synonymous codon usage

Alexander L. Cope [1,2,3]*, Premal Shah[1,2]*

**1** Department of Genetics, Rutgers University, Piscataway, New Jersey, United States of America, **2** Human Genetics Institute of New Jersey, Rutgers University, Piscataway, New Jersey, United States of America, **3** Robert Wood Johnson Medical School, Rutgers University, Piscataway, New Jersey, United States of America

\* alexander.cope@rutgers.edu (ALC); premal.shah@rutgers.edu (PS)

**Data Availability Statement:** All data are publicly available via the citations provided in this article (see doi:10.6084/m9.figshare. 5854692.v1). Scripts for re-creating our analysis and visualizations can be found at

## Abstract

Patterns of non-uniform usage of synonymous codons vary across genes in an organism and between species across all domains of life. This codon usage bias (CUB) is due to a combination of non-adaptive (e.g. mutation biases) and adaptive (e.g. natural selection for translation efficiency/accuracy) evolutionary forces. Most models quantify the effects of mutation bias and selection on CUB assuming uniform mutational and other non-adaptive forces across the genome. However, non-adaptive nucleotide biases can vary within a genome due to processes such as biased gene conversion (BGC), potentially obfuscating signals of selection on codon usage. Moreover, genome-wide estimates of non-adaptive nucleotide biases are lacking for non-model organisms. We combine an unsupervised learning method with a population genetics model of synonymous coding sequence evolution to assess the impact of intragenomic variation in non-adaptive nucleotide bias on quantification of natural selection on synonymous codon usage across 49 Saccharomycotina yeasts. We find that in the absence of *a priori* information, unsupervised learning can be used to identify genes evolving under different non-adaptive nucleotide biases. We find that the impact of intragenomic variation in non-adaptive nucleotide bias varies widely, even among closely-related species. We show that the overall strength and direction of translational selection can be underestimated by failing to account for intragenomic variation in non-adaptive nucleotide biases. Interestingly, genes falling into clusters identified by machine learning are also physically clustered across chromosomes. Our results indicate the need for more nuanced models of sequence evolution that systematically incorporate the effects of variable non-adaptive nucleotide biases on codon frequencies.

## Author summary

Codon usage bias (CUB), or the unequal usage of codons of the same amino acid (i.e. synonymous codons), has been observed in species across all domains of life. CUB is known to be shaped by both non-adaptive (e.g. mutation biases) and adaptive (e.g. natural

I[https://github.com/acope3/Intragenomic_variation_mutation_bias]. A table listing the accessions for RNA-seq data is provided in the Supplementary Material.

**Funding:** This work was supported by the National Science Foundation (DBI 1936046, https://www.nsf.gov/) and the National Institutes of Health (R35 GM124976, https://www.nih.gov/) awarded to P.S. A.L.C is currently supported by the INSPIRE (IRACDA New Jersey/New York for Science Partnerships in Research and Education) Postdoctoral Program (NIH PAR-19-366, https://www.nih.gov/). The funders had no role in study design, data collection and analysis, decision to publish, or preparation of the manuscript.

**Competing interests:** I have read the journal's policy and the authors of this manuscript have the following competing interests: P.S. is a Scientific Advisory Board member of Trestle Biosciences. P.S. also consults for Ribo-Therapeutics and an unnamed RNA-therapeutics startup.

selection for translation efficiency/accuracy) evolution. A key challenge for researchers is disentangling the role of various processes shaping codon usage, often for the purpose of identifying codons favored by natural selection, sometimes referred to as "optimal" or "preferred" codons. Despite large variation in non-adaptive nucleotide biases within a genome, most methods to quantify natural selection typically ignore this variation for the sake of simplicity. Here, we combine a population genetics model with unsupervised machine learning to identify genes evolving under different non-adaptive nucleotide biases across 49 budding yeasts species. We find that ignoring for variation in non-adaptive nucleotide biases can obfuscate signals of selection on codon usage. Our results indicate the need for more nuanced models of coding sequence evolution.

## Introduction

Patterns of nucleotide composition vary widely across a genome and are typically thought to be shaped by the interplay between adaptive and non-adaptive evolutionary processes. One common pattern observed in genomes across all domains of life is codon usage bias (CUB), the non-uniform usage of synonymous codons in coding sequences of genes [1–5]. Although synonymous nucleotide changes are often treated as neutral (i.e. codon frequencies are determined primarily by mutation bias and genetic drift) various lines of evidence indicate that synonymous changes are also subject to natural selection [6–12]. Coevolution between codon frequencies and the tRNA pool, as well as the bias towards translationally efficient codons in highly expressed genes suggests translational selection is a major factor shaping genome-wide codon patterns [8, 9, 13–19]. Other selective forces, including selection against missense error [10, 20], selection against ribosome drop-off [21, 22], and selection to avoid mRNA secondary structure near the translation initiation site [23, 24], are also hypothesized to shape adaptive CUB. Although evidence indicates strong purifying selection can act on synonymous changes [12, 25], codon usage is generally thought to be subject to weak selection (i.e. $N_e s \ll 1$) with genome-wide codon frequencies at selection-mutation-drift equilibrium [16, 26, 27].

Selection on codon usage related to mRNA translation is expected to be strongest in highly-expressed genes. Based on this assumption, various approaches for identifying the most efficient codon and quantifying selection on codon usage rely on comparing coding frequencies in highly-expressed genes to the remaining genome [15, 17, 19]. In contrast, other approaches quantify selection via the changes in codon frequencies as a function of gene expression, assuming that codon frequencies are at selection-mutation-drift equilibrium [16, 28, 29]. As selection on codon usage related to mRNA translation efficiency is expected to produce a correlation between CUB and gene expression, a weak correlation between codon usage and gene expression could indicate overall weak selection on codon usage.

Failing to account for mutational biases can weaken or completely obfuscate signals of selection on codons usage, resulting in the development of various methods for separating the effects of selection from mutation bias [16, 29–34]. Although current approaches often account for mutational biases when estimating selection on codon usage, these models often assume non-adaptive nucleotide biases (which included mutation bias) are constant across the genome. However, various processes cause the direction and strength of non-adaptive nucleotide biases to vary within the genome, potentially weakening signals of translational selection [17, 19, 35, 36]. Mutation rates can be context-dependent, varying based on the identity of adjacent nucleotides in bacteria [37], yeast [38], and primates and humans [39,

40]. Lateral gene transfer events, including introgressions, can result in genes with distinct CUB being incorporated into a genome [41, 42]. Mutation biases are also known to vary between leading and lagging DNA strands in some prokaryotic species, leading to different CUB dependent upon the strand of a coding sequence [43–45]. Notably, long stretches of DNA with relatively homogeneous nucleotide composition (e.g. GC-rich or GC-poor) are found in eukaryotic species ranging from yeasts to humans [46–49]. Various non-adaptive hypotheses are proposed to explain these long stretches of relatively GC-rich or GC-poor regions [46]. Gene conversion (BGC)—the transfer of genetic information from intact homologous sequences during repair of double-strand breaks [50]—has been observed to be GC-biased in various species. Previous work has hypothesized that gBGC could lead to the increased GC content in recombination hotspots and the correlation between recombination rates and GC content [17, 51–53]. Another potential hypotheses pertains to the timing of genome replication. As concentrations of free G and C nucleotides can vary during S-phase and this has been shown to impact misincorporation rates, mutation biases could arise as a result of replication timing [46, 54]. Finally, previous work has hypothesized that GC-rich and GC-poor regions could emerge due to a positive feedback loop between the rate of spontaneous deamination of C to T mutations and GC content [55, 56]. We know relatively little about the impact of variation in non-adaptive nucleotide biases on the relationship between codon usage and gene expression, and how this impacts estimates of selection on codon usage.

Population genetics models of coding sequence evolution are powerful tools for understanding the evolution of CUB [16, 28, 34]. We will use the Ribosomal Overhead Cost version of the Stochastic Evolutionary Model of Protein Production Rates (ROC-SEMPPR) to estimate codon-specific estimates of natural selection and mutation bias, as well gene-specific estimates of protein production rates [29]. Recent work applied ROC-SEMPPR to quantify the differences in codon usage between the ancestral genome and a large introgression in *Lachancea kluyveri*, finding that the ability to detect selection on codon usage improved if assuming codon usage in the introgressed region was shaped by different mutational and selective biases than the ancestral genes [42]. Here, we use ROC-SEMPPR to examine the effects of within-genome variation in mutation biases on the ability to detect natural selection on codon usage across the Saccharomycotina budding yeast subphylum [18]. Unlike *L. kluyveri*, *a priori* knowledge of genes evolving under different mutational biases is lacking in a large number of these species. To hypothesize genes shaped by different mutational biases, we apply an unsupervised machine-learning approach based on codon frequencies previously described in [57]. We highlight various yeasts in which intragenomic variation in non-adaptive nucleotide biases masks the efficacy of natural selection on codon usage. We find that genes falling into the same clusters determined via the unsupervised machine-learning algorithm are physically clustered within the genome. Although we cannot definitively comment on the biological causes for the intragenomic variation in non-adaptive nucleotide biases, our findings serve as the starting point for more directed studies.

## Results

To quantify the strength and direction of natural selection and mutation biases shaping codon usage patterns, we relied on the population genetics model ROC-SEMPPR as implemented in the AnaCoDa R package [29, 58]. Briefly, ROC-SEMPPR is a Bayesian model that estimates natural selection and mutation bias by accounting for the expected covariation between observed codon counts (taken from protein-coding sequences) and gene expression (see Methods for more details). Briefly, for any amino acid with $n_{aa}$ synonymous codons, the

probability of observing codon $i$ in gene $g$ can be described by the equation

$$p_{i,g} = \frac{e^{-\Delta M_i - \Delta \eta_i \phi_g}}{\sum_j^{n_{aa}} e^{-\Delta M_j - \Delta \eta_j \phi_g}}$$

where $\Delta M_i$ and $\Delta \eta_i$ represent mutation bias and selection coefficient of codon $i$ relative to a reference synonymous codon, and $\phi_g$ represents gene expression of gene $g$. In the absence of empirical gene expression data, ROC-SEMPPR can estimate an evolutionary average value of gene expression from observed codon counts by assuming gene expression follows a lognormal distribution, allowing us to apply the framework to even species lacking expression data. Parameters are estimated via a Markov Chain Monte Carlo (MCMC) simulation approach. The current implementation of ROC-SEMPPR allows users to specify sets of coding sequences that are evolving under different codon-specific selection coefficients and/or mutation biases. This framework has previously been used to examine the impact of a large introgression on quantifying codon usage in *L. kluyveri* by assuming coding sequences in the introgressed region were evolving under a different set of codon-specific parameters [42].

## Relationship between observed and predicted gene expression can vary significantly between closely-related species

We applied ROC-SEMPPR to the nuclear protein-coding genes of 49 species from the Saccharomycotina budding yeast subphylum for which we were able to obtain empirical estimates of mRNA abundances via RNA-seq (see Methods). We will refer to these fits as the constant mutation ("ConstMut") model as mutation bias parameters are assumed to be the same across all coding sequences. Overall, we find that predicted expression levels from ROC-SEMPPR (based solely on codon usage frequencies and the assumption that expression levels are lognormally distributed) are correlated with empirical mRNA abundance estimates from RNA-seq data (S1 Fig).

Although ROC-SEMPPR sufficiently estimates selection on codon for the majority of species, there are some species for which the correlation between predicted and empirical gene expression estimates appears significantly weaker compared to their closely-related sister taxa. We use closely-related species across three genera (*Saccharomyces*, *Candida*, and *Ogataea*) to highlight the differences in the relationship between empirical gene-expression levels and the evolutionary average expression levels predicted based on codon usage patterns using ROC-SEMPPR. For each of the three genera, one of the species shows a relatively high correlation between empirical and predicted expression levels, while the other species in the same genus show weak or even negative relationships (Fig 1A). These discrepancies might be due to three reasons—(i) poor quality of expression data for these non-model organisms, (ii) rapid changes in the degree of translation selection across closely related species, or (iii) inaccurate predictions of gene-expression data from CUB due to model misspecification.

When comparing expression data between closely-related species, we find they are of similar sequencing depth and are moderately correlated with each other (S2 Fig). This indicates that variation in the quality of the RNA-seq datasets is insufficient to explain the large changes in correlation between observed and predicted expression levels across closely-related species. To examine if the degree and direction of translation selection on codon usage differ between closely-related species of the three genera, we compared the relative synonymous codon usage (RSCU) between genes with high and low expression levels (top 5% and bottom 5%, respectively). For reference, RSCU compares observed codon frequencies relative to the expected frequencies assuming synonymous codon usage is unbiased [59]. Across all three genera, RSCU values for each codon calculated from highly-expressed genes are well-correlated, indicating

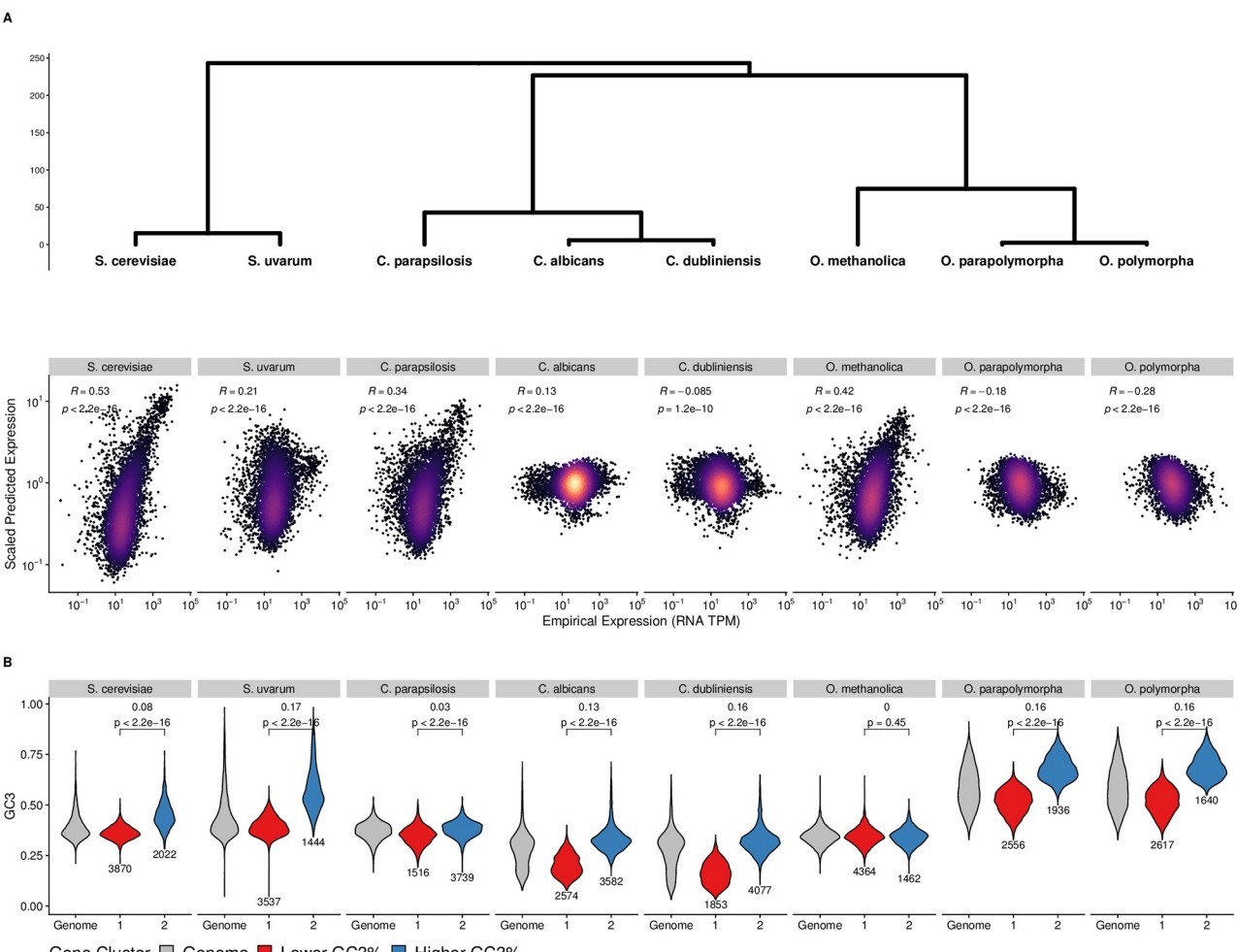

**Fig 1. Ability to detect selection on codon usage varies between closely-related species.** (A) Correlations between empirical mRNA abundances and predicted gene expression values for all genes from ROC-SEMPPR across 8 Saccharomycotina species. Phylogenetic relationships between species indicate relatedness between species, with branch lengths units in millions of years. (B) Distribution of GC3 for the entire genome (grey), as well as for the distributions within each of the clusters determined by the CLARA clustering. We denote the cluster with the lower median GC3 as the Lower GC3% cluster (red) and the cluster with the higher median GC3% as the Higher GC3% cluster (blue). Brackets indicate the difference in medians between the lower and higher GC3% clusters, as well as the p-value from a Kolmogorov-Smirnov test comparing the distribution of GC3% between the two clusters.

that the overall direction of selection on synonymous codon usage is similar between species (S3 Fig). For all 8 species, RSCU values from low and high expression genes are correlated, but the RSCU values of many codons differ between high and low expression genes (S4 Fig). Taken together, these results suggest the strength and direction of selection on synonymous codons are similar across species within the same genus.

To test if the poor correlation between observed and predicted expression data are unique to ROC-SEMPPR, we also estimated Codon Adaptation Index (CAI) [60] values for genes in the above species. Although ribosomal proteins are often used as a reference set when calculating CAI, in many cases, they may not serve as an appropriate reference set [61]. As a result, we estimated CAI using the automatic detection of a reference set via correspondence analysis in CodonW [62] (S5 Fig). A key assumption of the CodonW approach, much like ROC-SEMPPR, is the assumption that the primary driver of codon usage variation across genes is

gene expression. We find that predicting expression using CAI leads to similar results as ROC-SEMPPR for these species (S5 Fig), indicating that models underlying both CAI and ROC-SEMPPR are misspecified.

## Intragenomic variation in non-adaptive nucleotide biases affects estimates of translational selection on codon usage

Models of codon usage evolution typically assume that the non-adaptive nucleotide biases associated with a particular synonymous codon are constant across the entire genome. However, several non-adaptive forces can affect the local mutation rates and nucleotide biases and, therefore, affect codon composition. As many of the yeasts of interest are non-model species, relevant data, such as recombination rates for assessing the impact of biased gene conversion, are lacking.

To hypothesize protein-coding sequences within each species that may be evolving under locally variable non-adaptive nucleotide biases, we assigned coding sequences to one of two clusters based on the k-medoids CLARA clustering algorithm applied to the results from correspondence analysis of codon frequencies (see Methods for details). This approach has been previously used to identify laterally transferred genes in *E. coli* [57, 63]. As evidence for the validity of this clustering approach, we note that one of the mutation regimes identified in *L. kluyveri* predominantly consisted of coding sequences found in the introgressed region on Chromosome C (S6 Fig, SAKL0C, blue bars).

Interestingly, we find that genes in the two clusters differed in their GC3% content (Fig 1B), suggesting some genes may be subject to stronger GC nucleotide biases than others. A clear pattern emerges when examining the differences in GC3% content between the Lower GC3% and Higher GC3% cluster: species with larger differences (i.e. > 0.1) in the median GC3% content between clusters also have weaker or negative correlations between predicted and empirical gene expression (Fig 1B and 1C). We note that the distributions of GC3% between clusters for each species, with the exception of *O. methanolica*, are significantly different (Kolmogorov-Smirnov test, $p < 2.2e - 16$). This indicates that while the CLARA clustering algorithm may be using GC3 as one of the discriminating factors between genes, it is not the sole feature for clustering.

We now fit a modified ROC-SEMPPR model to all species, where we allow mutation bias parameters to vary between the two sets of coding sequences determined by the clustering algorithm. We still assume that codon-specific selection coefficients are the same for genes in both clusters because they share the same cytoplasm and are constrained by the same tRNA pool. We will refer to these model fits as the varying mutation ("VarMut") model. We note that even in cases where non-mutational processes, such as biased gene conversion, are driving non-adaptive nucleotide biases, we expect these sources of bias will be absorbed into ROC-SEMPPR's mutation bias parameters because these nucleotide biases are not expected to covary with gene expression.

Species showing greater differences in GC3% between the clusters showed significant improvement in the correlation between predicted and empirical gene expression when accounting for intragenomic variation in mutation bias (Fig 2A and 2B). For species with relatively little difference in GC3% between clusters, the VarMut model often resulted in a poorer agreement between empirical and predicted expression estimates. This suggests that clustering in cases where intragenomic variation in mutation bias is small or non-existent dilutes signals of selection on codon usage. In the 5 species for which the VarMut model fit significantly improved predictions of gene expression, the predicted expression estimates from the ConstMut model fit essentially capture the Lower GC3% and Higher GC3% clusters (Fig 2C).

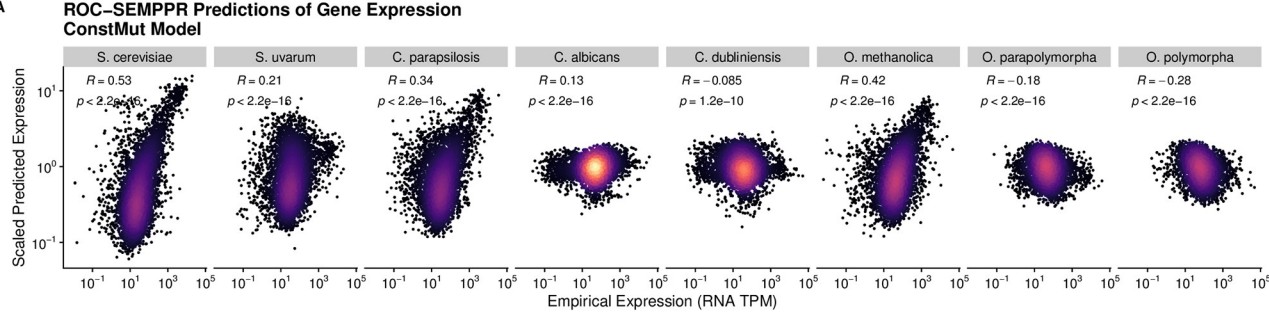

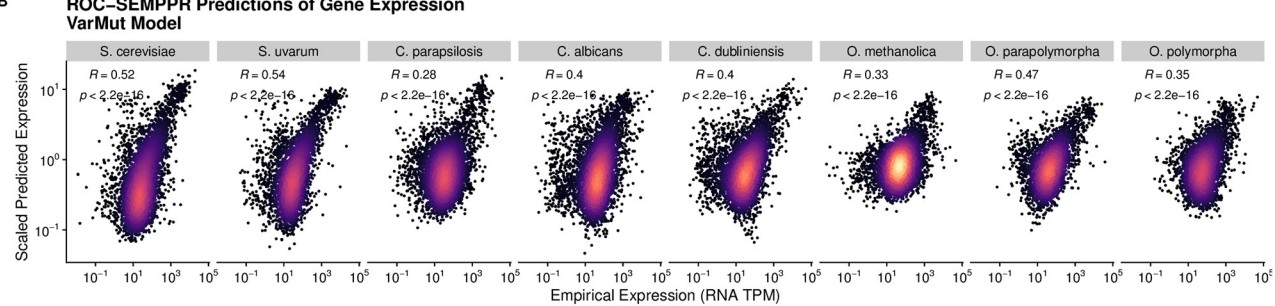

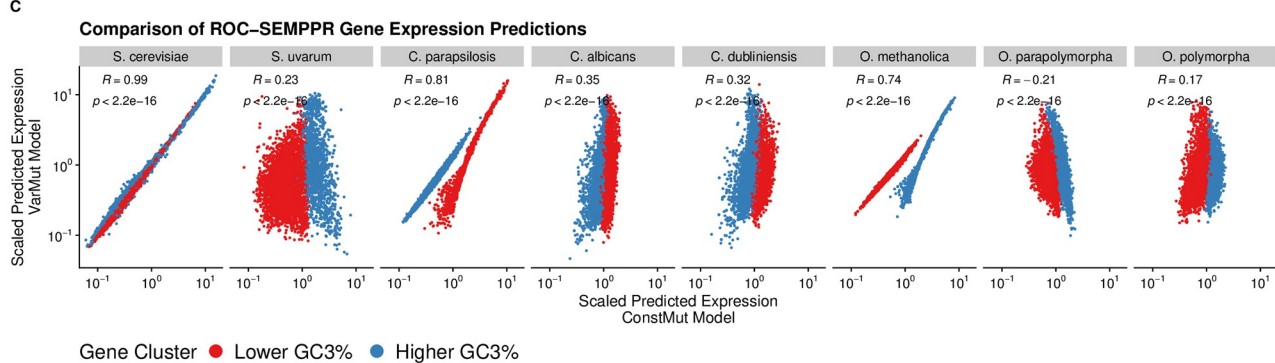

Gene Cluster ● Lower GC3% ● Higher GC3%

**Fig 2. Effects of allowing for non-adaptive nucleotide biases to vary across genes on predictions of gene expression.** (A) Correlation between empirical and predicted gene expression values for each gene when mutation bias parameters are shared across clusters (ConstMut Model), as in Fig 1. (B) Correlation between empirical and predicted gene expression values when mutation bias parameters are allowed to vary across clusters (VarMut model). (C) Comparison of predicted gene expression between the ConstMut and VarMut models.

Importantly, non-adaptive nucleotide biases can impact codon usage regardless of gene expression, whereas selection on codon usage is generally strongest in a subset of highly-expressed genes. As ROC-SEMPPR assumes that the primary driver of covariation in codon usage is gene expression, our results suggest that variation in non-adaptive nucleotide biases, when strong enough, can be mistaken for natural selection on codon usage.

The above analyses focused on partitioning genes into two clusters. However, it remains unclear if two is the ideal number of clusters or whether increasing the number of clusters will further improve our ability to predict expression levels and more fully capture the patterns of codon usage. To address this, we started by partitioning genes for all five species considered above into three clusters. As can be seen in S7 Fig, 3 clusters usually resulted in poor(er) ability to predict gene expression from codon usage patterns than using 2 clusters in most cases. This

is perhaps not surprising in hindsight because with a higher number of clusters, we run the risk of over-partitioning genes such that it is likely that most or all highly expressed genes might get clustered into a single group. As a result, our ability to separate the effects of non-adaptive and adaptive forces on codon usage patterns using ROC-SEMPPR diminishes significantly.

## Genes in a mutational cluster are physically clustered on chromosomes

Our results indicate that differences in non-adaptive nucleotide biases across protein-coding sequences mask signals of selection on codon usage in species showing greater variability in GC3%. To gain insight into the potential mechanisms responsible for the clustering of genes in different GC3% categories, we checked if genes belonging to individual clusters are in close proximity to each other on the chromosome. Many mechanisms could hypothetically lead to clusters of GC-rich or GC-poor genes, such as biased gene conversion or the availability of free GC nucleotides during DNA replication. To quantify the degree of physical clustering of genes belonging to each mutational cluster, we estimated GC3% variation along chromosomes using either a 20-gene moving average approach or 20-gene non-overlapping windows.

We observe that genes assigned to the same mutational cluster are also physically clustered across the chromosomes of species for which the VarMut model was better able to detect selection on codon usage relative to the ConstMut model (Fig 3 and S8–S15 Figs). As with the mutational clusters (Fig 1C), physical clusters of genes assigned to the same mutational cluster reflect relatively large (sometimes spanning hundreds of genes) GC3-rich or GC3-poor regions (Fig 3, left panels). Using non-overlapping 20-gene windows, which allows us to apply statistics that assume independent data, we observe that the average GC3% within a region is highly correlated with the percentage of genes assigned to the Higher GC3% within that region across all chromosomes of a species (Fig 3, right panels). However, physical clustering of genes assigned to the same mutational cluster was overall weaker across species for which the VarMut model failed to improve the ability to detect selection on codon usage (S16 Fig).

## Effects of variation in non-adaptive nucleotide biases on estimates of natural selection

Allowing for intragenomic variation in non-adaptive nucleotide biases improved detection of natural selection on codon usage. A key aspect of studying CUB is identifying the "optimal" or "preferred" codon. Across the 5 species for which we found the VarMut model outperformed the ConstMut model, the optimal codon was misidentified for the majority of amino acids by the latter (Fig 4B). In the case of amino acids for which the optimal codon was the same between models, selection coefficient estimates for many codons showed large changes between the models (Fig 4C). This indicates that the strength of selection may be over or under-estimated when various sources of non-adaptive nucleotide biases are unaccounted for, even if the optimal codon is correctly identified.

Selection coefficient estimates from ROC-SEMPPR represent the preference of a codon relative to a single reference codon. As the clusters reflect differences in GC3%, selection coefficients were modified to represent the strength and direction selection on NNG codons relative to NNA codons, and NNC codons relative to NNT codons. GC-ending codons are universally disfavored by natural selection relative to AT-ending codons in *C. albicans* when fitting the ConstMut model (S17 Fig). This is consistent with the distributions of predicted expression estimates, with genes falling into the Higher GC3% cluster predicted to have lower expression

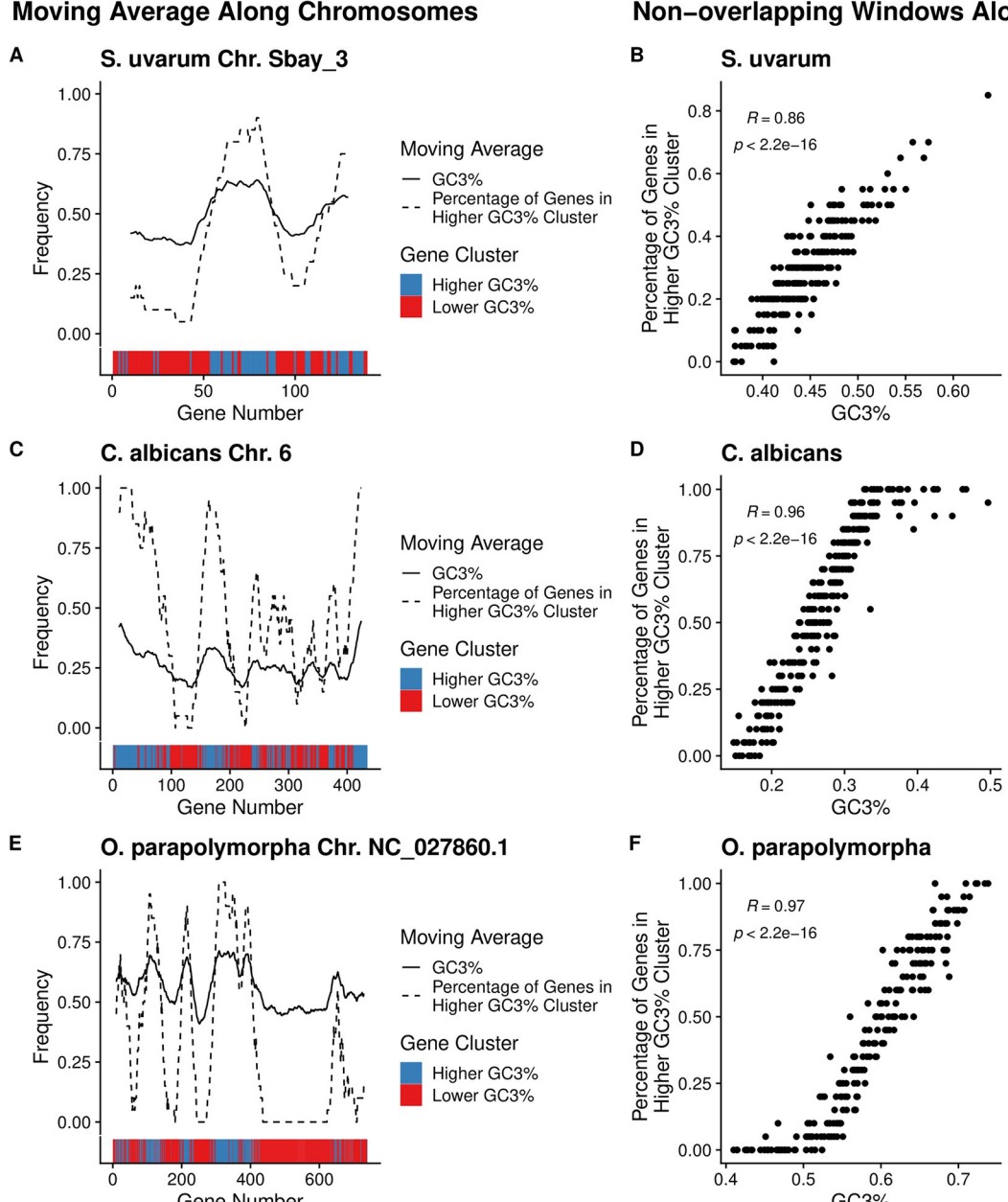

**Fig 3. Genes evolving under similar non-adaptive nucleotide biases are cluster along chromosomes.** (Left panel) Per-gene GC3% content across select chromosomes quantified as a moving average using a 20 gene sliding window (solid line). For each 20 gene window, the percentage of genes assigned to the Higher GC3% regime is also shown (dashed line). Color bars indicate the mutation regimes for Higher and Lower GC3% (blue and right, respectively). (Right panel) Scatter plots showing per-gene GC3% and percentage of genes assigned to the Higher GC3% regime using non-overlapping 20 gene windows across all chromosomes. The independence of windows allows us to apply the Spearman Rank correlation. (A,B) *S. uvarum*. (C,D) *C. albicans*. (E,F) *O. parapolymorpha*.

(Fig 2C). The VarMut model reveals that some GC-ending codons are favored by natural selection relative to the corresponding AT-ending codon (S17 Fig). Even in cases where the GC-ending codon remains disfavored relative to the AT-ending codon, we find that the magnitude of the estimated selection coefficients is lower in the VarMut model (S17 Fig), indicating selection against these codons is weaker than suggested by the ConstMut model. Other

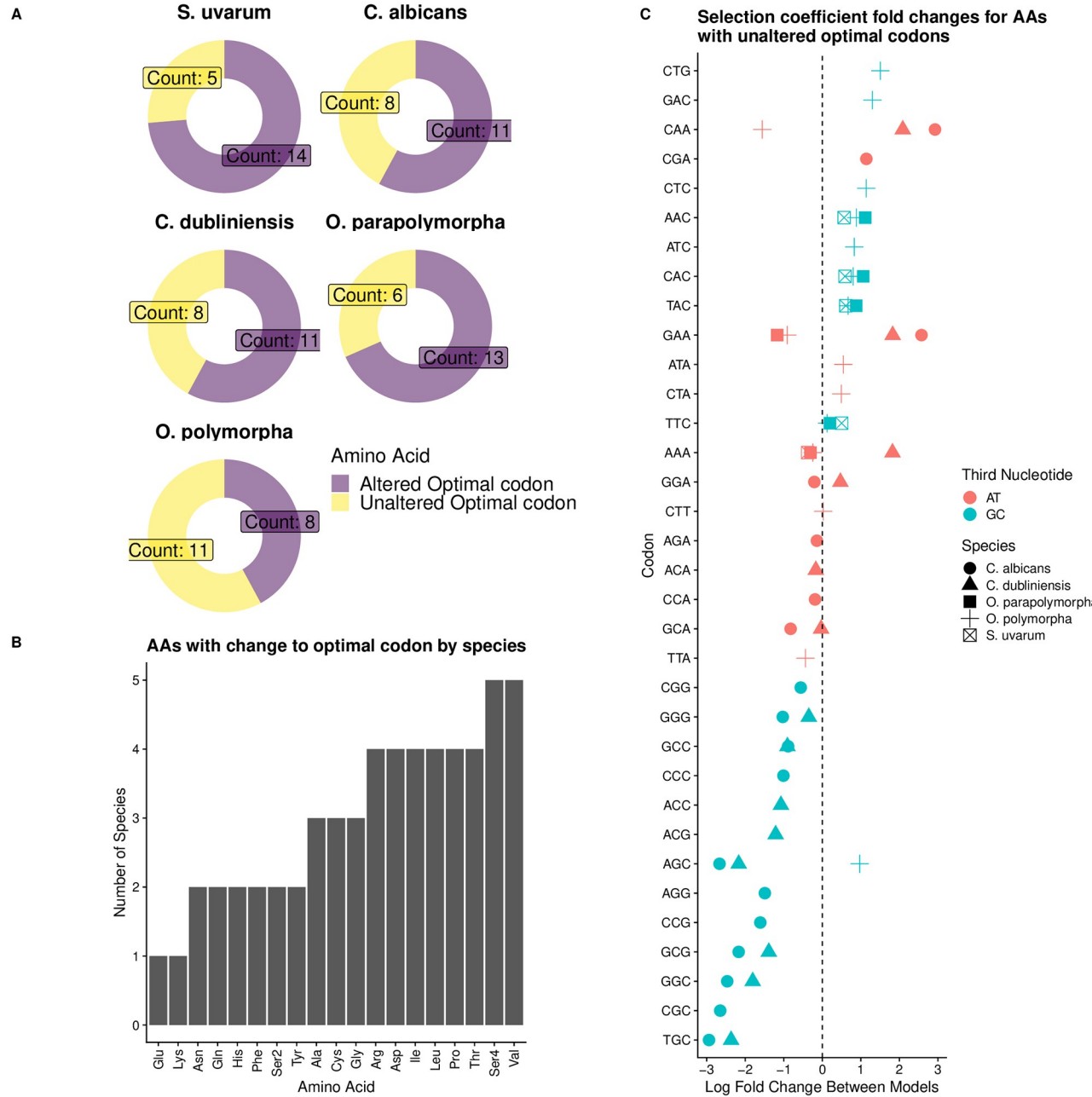

**Fig 4. Failing to account for variation in non-adaptive nucleotide biases can lead to mis-identification of optimal codons.** Data represent results from *S. uvarum*, *C. albicans*, *C. dubliniensis*, *O. parapolymorpha*, and *O. polymorpha*. (A) Number of amino acids for which the optimal codon differs between the ConstMut and VarMut models. (B) Number of species per amino acid that had a change in the identified optimal codon between the VarMut and ConstMut models. Ser4 indicates serine codons of the form TCN, where N is any nucleotide, while Ser2 indicates those of the form AGY, where Y is C or T. (C) Breakdown of fold changes for amino acids in which optimal codon is the same between ConstMut and VarMut models. Shapes indicate species, while color indicates if the third nucleotide is A/T or G/C.

species for which the VarMut model improved ROC-SEMPPR's ability to detect natural selection on codon usage showed similar results (S17 Fig). Unsurprisingly, we find that mutation bias estimates against GC-ending codons (again, relative to AT-ending codons) are much weaker in the Higher GC3% cluster than the Lower GC3% cluster in these species (S18 Fig).

## Detection of selection on CUB improves when allowing for non-adaptive nucleotide bias variation in species across the Saccharomycotina subphylum

Up to this point, our analysis of the effects of intragenomic variation in non-adaptive nucleotide biases has primarily focused on 8 species represented by three genera within the Saccharomycontina subphylum. To understand the effects of intragenomic variation in non-adaptive nucleotide biases across the larger phylogeny, we return our focus to the 49 species across the subphylum for which reliable expression datasets are available (Fig 5A). Accounting for intragenomic non-adaptive nucleotide bias variation has a relatively small impact on our ability to detect selection on codon usage for the majority of species (Fig 5A). However, the correlations between empirical and predicted genes expression estimates improve in various species, suggesting that nucleotide biases are shaped by multiple non-adaptive processes across the Saccharomycontina subphylum. We find that species with larger variation in GC3% content, quantified as the difference in median GC3% content between the Lower and Higher GC3% clusters, tend to show larger improvements in the correlations between empirical and predicted expression estimates when fitting the VarMut Model (Spearman Rank correlation

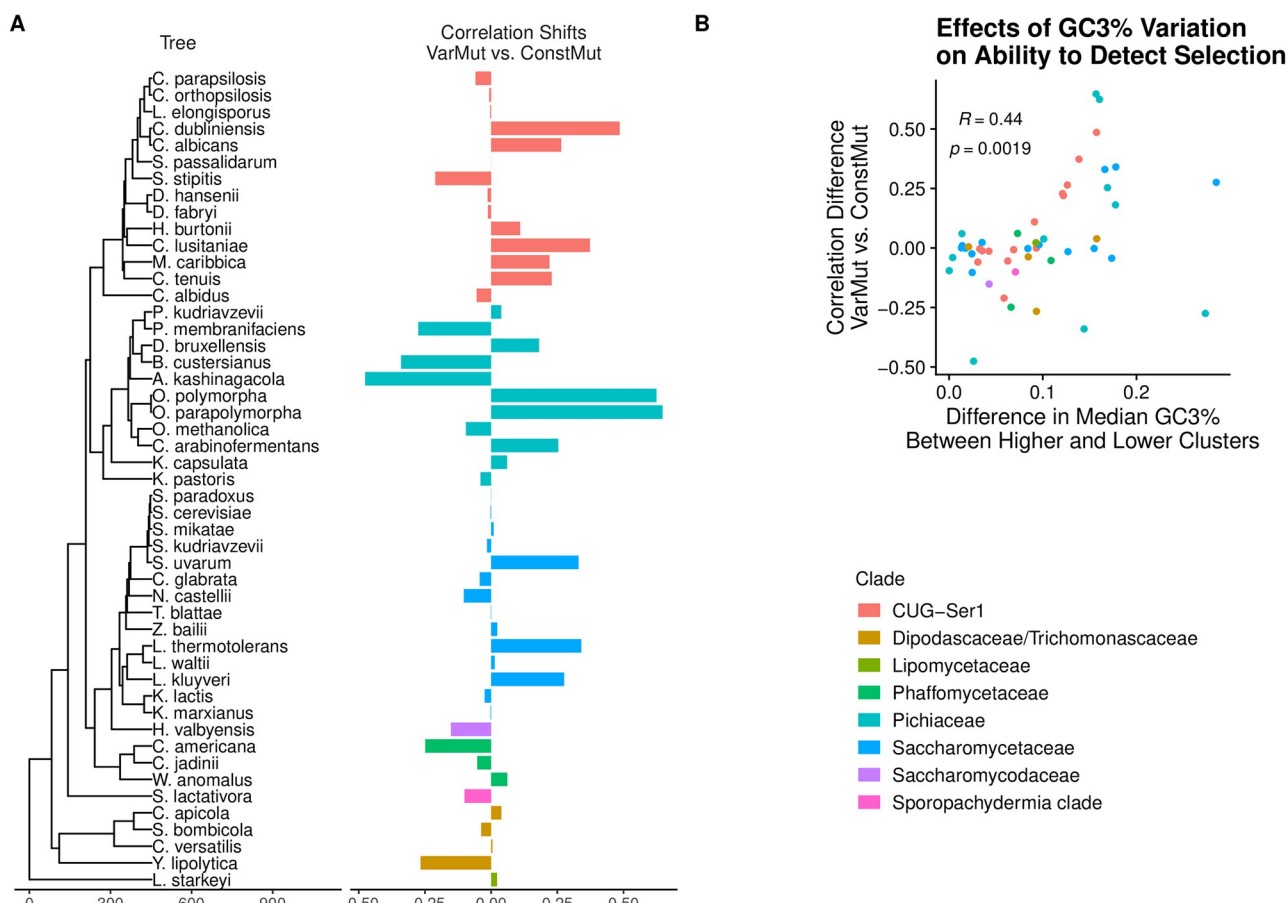

**Fig 5. Impact of intragenomic variation in non-adaptive nucleotide biases on predicting gene expression across 49 Saccharomycotina yeasts.** Clade names are taken from [18]. (A) Change in Spearman correlation between empirical and predicted gene expression when fitting VarMut model relative to ConstMut model. A positive value indicates an improvement in correlation when fitting the VarMut model. (B) Relationship between the shift in correlations as a function of the difference in median GC3% content between the Lower and Higher GC3% clusters.

$r = 0.44, p = 0.0019$, Fig 5B). This correlation remains statistically significant after transforming values using phylogenetic independent contrasts (Spearman Rank Correlation $r = 0.48$, $p = 0.0006$) [64].

Interestingly, the effect of intragenomic variation in non-adaptive nucleotide biases can vary between closely related species, as previously highlighted in species from the *Saccharomyces*, *Candida*, and *Ogataea* genera. In addition, the causes of this variation may also vary between closely-related species. For example, both *L. kluyveri* and *L. thermotolerans* show a substantial improvement in the predictions of gene expression with the VarMut model; however, only the former contains the large introgressed region on one of its chromosomes [65–67]. In the case of *L. kluyveri*, the Higher GC3% cluster is predominantly the large introgressed region found on Chromosome C (S6 Fig). In contrast, *L. thermotolerans* shows relatively large variation in GC3% content along its chromosomes (S19 Fig), similar to *C. albicans* and *C. dubliniensis*, but the mechanistic basis of this variation is unclear.

As an alternative method for evaluating the relative performance of the ConstMut and VarMut models, we compared the Deviance Information Criterion (DIC), a Bayesian generalization of the Akaike Information Criterion, for both model fits [68]. Consistent with expectations, we see that the ΔDIC (i.e. $DIC_{VarMut} - DIC_{ConstMut}$) decreases on average as the the VarMut model improves the predictions of gene expression (note: models with lower DIC scores are considered better). Surprisingly, the VarMut model is often favored over the ConstMut model based on DIC scores, even in cases where the VarMut model makes worse predictions of gene expression (S20 Fig). Generally, the larger the difference in GC3% observed between the two clusters, the more favored the VarMut model appears to be (S20 Fig).

## Discussion

A key to understanding the evolution of codon usage bias (CUB) across a phylogeny is to quantify the relative contributions of non-adaptive (e.g. mutation bias) and adaptive evolutionary forces in shaping codon frequencies. Studies often assume that non-adaptive nucleotide biases are constant across the genome; however, this may not always be the case due to various processes, such as biased gene conversion (BGC). Across 49 Saccharomycontina yeasts, failing to account for intragenomic variation in non-adaptive nucleotide biases obfuscates signals of natural selection on codon usage in various phylogenetically-diverse species. This includes the misidentification of the direction of selection (i.e. misidentification of optimal codons) and the over/under-estimation of the strength of selection. In the absence of *a priori* hypotheses, such as large introgression events [42], unsupervised clustering of genes based on codon frequencies can reveal genes evolving under different non-adaptive nucleotide biases. Interestingly, we observe in many species that the unsupervised clustering is mirrored by physical clustering along chromosomes, i.e. genes falling into the same mutational cluster tend to be physically grouped within the genome. For some species, such as *C. albicans* and *O. parapolymorpha*, this is a constant across chromosomes, with each chromosome showing large variations in GC3% content. For other species, such as *S. uvaraum*, regions of large variation in GC3% content appear to be more isolated. Species showing little variation in GC3% content across their chromosomes showed little, if any improvement, by allowing for intragenomic variation in non-adaptive nucleotide biases.

To categorize genes evolving under different non-adaptive nucleotide biases, we used the CLARA clustering algorithm (a type of k-medioids clustering) and clustered genes into two groups ($k = 2$). We note that using $k = 3$ clusters did not appear to perform much better than ConstMut model fits for the 5 species highlighted in Fig 1 (S7 Fig) and in most cases performed significantly worse. It is possible that identifying more clusters results in further

improvements in the detection of natural selection on codon usage, but we do not currently have examples of this. It is important to remember that for most species under strong translational selection on codon usage, a major factor shaping variation in codon frequencies is gene expression. As ROC-SEMPPR estimates parameters by accounting for the expected covariation between codon usage and gene expression, over-clustering of genes could potentially dilute signals of natural selection. We find evidence of this in species that have low variation in GC3% along the genome, such as *C. parapsilosis*, in which the VarMut model performs significantly worse at predicting gene expression.

The physical clustering of genes along a chromosome that are assigned to either the Higher or Lower GC3% clusters is consistent with a variety of biological processes. The presence of GC-rich and GC-poor regions within a genome is hypothesized to be due to GC-biased gene conversion (gBGC), mutation biases related to variability in free nucleotide concentration during S-phase, and/or feedback between the rate of spontaneous deamination of C to T and GC content [46, 52, 54, 55]. gBGC is a popular hypothesis for explaining variations in GC% content within eukaryotic genomes. Assuming recombination rates are well-conserved across species, [17] concluded gBGC had the strongest impact on codon usage in *S. uvarum* (*S. bayanus*), which was the only *sensu stricto* yeast for which the VarMut model did a better job of predicting gene expression than the ConstMut model. The large variations of GC content in *C. albicans* and *C. dubliniensis* are theoretically consistent with either gBGC via mitotic recombination (as neither species reproduces sexually [69, 70]) or variations in nucleotide concentrations during genome replication [49], but we currently lack empirical validation. Although gBGC has long been thought to occur in *S. cerevisiae* and closely-related species, recent work by [71] found no evidence to support biased gene conversion in *S. cerevisiae* and hypothesizes that recombination may occur more frequently in GC-rich regions, consistent with the findings of [72]. Higher recombination rates <u>caused</u> by higher GC content could also explain the correlation between GC% and recombination rates, as seen across the *sensu stricto* yeasts in [17]. Given the results of [71] and the lack of recombination rate estimates across the Saccharomycotina subphylum, we cannot confidently attribute our results to gBGC in these species. Our results do not serve as evidence (nor are they meant to) for any of the hypotheses explaining within-genome variation in GC content, which are not necessarily mutually-exclusive in generating regions of high and low GC content. We encourage researchers to use our results as a starting point for more targeted theoretical and empirical work trying to understand the within-species and across-species evolution of GC content across the Saccharomycontina subphylum.

As expected, differences in the Deviance Information Criterion (DIC) were anti-correlated with the differences in the correlation between predicted and empirical gene expression estimates when comparing the VarMut and ConstMut models. In other words, the better the VarMut model did at predicting gene expression compared to the ConstMut model, the more preferred it was according to model comparisons via DIC. Surprisingly, model comparisons via DIC predominantly favored the VarMut model, even in cases where this model did worse at predicting gene expression. This could be, in part, due to the tendency for DIC to favor overfit models [73]. However, this also highlights a potential limitation of coupling ROC-SEMPPR with unsupervised learning. The clustering of genes will attempt to maximize differences in codon usage patterns, even if these differences do not represent significant differences in mutation biases and/or natural selection. As ROC-SEMPPR works solely with codon count data and attempts to estimate gene expression assuming a lognormal distribution, the lack of grounding in empirical information could allow ROC-SEMPPR to reach parameter estimates that reasonably reflect codon counts, but may not have empirical justification. Previously, [74] found that information criterion could favor incorrect models when the data violated the

models' assumptions, although it is unclear if over-clustering of genes could violate ROC-SEMPPR's assumptions.

Tests of selection on codon usage are often based on comparing the codon usage of a gene to a reference set thought to be biased towards selectively-favored codons. These tests often create a reference set using assumed highly expressed genes (e.g. ribosomal proteins [75], but see [61]) or using empirical gene expression data [19]. In the case of intragenomic variation in non-adaptive nucleotide biases, the reference set must be chosen to minimize the impact of these various biases, as this may result in an over or underestimation of the strength of selection. We found that using data-driven approaches to identifying a reference set using CodonW resulted in poor performance of CAI, similar to ROC-SEMPPR ConstMut fits. Even if care is taken in choosing the reference set, variation in non-adaptive nucleotide biases may still lead to the over or underestimation of selection on codon usage in specific genes. For example, if selection generally favors GC-ending codons, then natural selection on codon usage may be overestimated in low expression genes subject to gBGC.

Approaches for dealing with variation in non-adaptive nucleotide composition biases rely on the analysis of SNPs detected across multiple individuals from a population, or comparisons of codon and nucleotide frequencies in regions of low and high recombination [17, 35, 36, 76, 77]. However, polymorphism and recombination data are often unavailable for non-model species, limiting the applicability of these approaches. We combined unsupervised machine-learning with an explicit populations genetics model to estimate selection on codon usage in the context of variable non-adaptive nucleotide biases. Although machine-learning is a powerful tool, the descriptive nature of such approaches can make biological interpretations difficult. As our understanding of the coding sequence evolution matures, models that more explicitly incorporate the various evolutionary forces that shape codon usage patterns are necessary.

## Materials and methods

We obtained genome sequences and associated annotation files from [78]. We excluded mitochondrial genes, protein-coding genes with non-canonical start codons, internal stop codons, and sequences whose lengths were not a multiple of three from all analyses. To identify mitochondrial genes, we queried all protein sequences against a BLAST database built from sequences in the MiToFun database (http://mitofun.biol.uoa.gr/).

Empirical mRNA abundances were used as a proxy of protein production rates. For each species, publicly available RNA-seq data were downloaded from the Sequence Read Archive or the European Nucleotide Archive (*L. kluyveri only*) (S1 Table). Adapters for each sequence were trimmed using fastp [79] and genes were quantified using kallisto [80]. Transcripts-per-million (TPMs) were re-calculated for each transcript by rounding raw read counts to the nearest whole-number [81].

### Identifying intragenomic variation in codon usage bias

To identify genes potentially evolving under different non-adaptive nucleotide biases, an unsupervised learning approach was implemented as described in [57]. Correspondence Analysis (CA) is a multivariate technique similar to principal component analysis that works on categorical data. We performed correspondence analysis for each species based on 61 sense codon frequencies using the *ca* R package [82]. In cases where selection on codon usage is the primary driver of variation in codon usage across the genes, the first principle component of CA often correlates with gene expression [57]. We note that Relative Synonymous Codon Usage (RSCU) [59], which represents the observed frequency of a codon relative to the expectation

assuming synonymous codon usage is unbiased, was not used as it has been found to introduce artifacts to CA [83]. Genes were then clustered into two groups based on the first 4 principal components from the CA using the CLARA algorithm implemented in the cluster R package [84]. The CLARA algorithm is designed to perform k-medoids clustering on large datasets, where k-medoids centers clusters on actual data points, in contrast to k-means, which centers clusters based on the average of the data points in the cluster. The CLARA algorithm first generates a user-specific number of subsets of the data, each of a user-specified size (in our case, half of the protein-coding sequences for each species were used). The medoids are identified for each subset and then all observations (in our case, all protein-coding sequences for a species) are assigned to a medoid based on a measure of distance (in our case, Euclidean distance). The best overall set of medoids is chosen based on the mean dissimilarities between each observation and its assigned to a medoid. For each species, the cluster with the lower median GC3% was designated as "Lower GC3% Cluster" and the cluster with the higher median GC3% was designated as the "Higher GC3% Cluster". We also tested clustering genes into three groups, but found these results to generally be worse than when using two groups.

## ROC-SEMPPR model of codon usage evolution

Ribosomal Overhead Cost version of the Stochastic Evolutionary Model of Protein Production Rates (ROC-SEMPPR) was independently fit to 49 species using the R package AnaCoDa with the number of protein-coding sequences included ranging from 4,200 to 8,700, depending on the species [58]. ROC-SEMPPR is implemented in a Bayesian framework, allowing it to simultaneously estimate codon-specific estimates of selection coefficients and mutation bias with gene-specific estimates of the evolutionary average gene expression by assuming gene expression is lognormally distributed [29]. This allows the model to be fit to any species with annotated protein-coding sequences. Included again for simplicity, for any amino acid with $n_{aa}$ synonymous codons, the probability of observing codon $i$ in gene $g$ can be described by the equation

$$p_{i,g} = \frac{e^{-\Delta M_i - \Delta \eta_i \phi_g}}{\sum_j^{n_{aa}} e^{-\Delta M_j - \Delta \eta_j \phi_g}} \tag{1}$$

where $\Delta M_i$ and $\Delta \eta_i$ represent mutation bias and selection coefficient of codon $i$ relative to a reference synonymous codon, and $\phi_g$ represents gene expression of gene $g$ which follows from the steady-state distribution of fixed genotypes under selection-mutation-drift equilibrium [16, 29, 85]. In the absence of empirical gene expression data, ROC-SEMPPR can estimate an evolutionary average value of gene expression from observed codon counts by assuming gene expression follows a lognormal distribution, allowing us to apply the framework to even species lacking expression data. Note that selection-mutation-drift equilibrium assumes that selection on codon usage is generally weak ($N_e s \ll 1$, where $N_e$ is the effective population size and $s$ is the selection coefficient). As is standard when using the Greek letter $\Delta$, the codon-specific parameters reflect the strength and direction of selection and mutation relative to another synonymous codon, arbitrarily chosen as the alphabetically last codon. For each gene, the observed codon counts for an amino acid are expected to follow a multinomial distribution with the probability of observing a codon defined by Eq 1. Given the codon counts and the assumption that gene expression follows a lognormal distribution, ROC-SEMPPR estimates the parameters that best fit the codon counts via a Markov Chain Monte Carlo simulation approach (MCMC). For each species, MCMC were fit for 200,000 iterations, keeping every 10[th] iteration. The first 50,000 iterations were discarded as burn-in. For each analysis, two separate chains were run from different starting points and parameter estimates were compared

to assess convergence. As ROC-SEMPPR is a Bayesian model with parameters estimated via an MCMC, model comparisons were performed using the Deviance Information Criterion (DIC), which is a Bayesian generalization of the Akaike Information Criterion [68]. Aside from being more appropriate than AIC for Bayesian analyses, DIC is also readily calculated from MCMC samples.

Previous work with ROC-SEMPPR has typically separated serine codons TCN (where N is any of the other 4 nucleotides) and AGY (where Y is C or T) into separate groups of codons for the analysis [16, 29, 42]. ROC-SEMPPR assumes weak-mutation (i.e. $N_e\mu \ll 1$, where $\mu$ is the mutation rate) such that each mutation introduced to a population is fixed or lost before the arrival of the next mutation. ROC-SEMPPR also assumes a constant amino acid sequence, such that going between these two groups of serine codons would require the fixation of a non-serine amino acid before returning to serine via the fixation of another mutation, clearly violating the fixed amino acid sequence assumption. To handle species for which CTG codes for serine instead of the canonical amino acid leucine, a local version of AnaCoDa was created that is capable of handling this amino acid switch. For these species, CTG was also treated as a third codon group for serine, similar to ATG (methionine) or TGG (tryptophan), which have no synonyms.

For each species, ROC-SEMPPR was first fit to the protein-coding sequences of each species assuming selection coefficient and mutation bias parameters were the same between the two clusters, which we refer to as the "ConstMut" model. Similarly, the protein-coding sequences of each species were also fit allowing the mutation bias to vary for genes indicated by the two clusters, which we will refer to as the "VarMut" model. For the VarMut model, selection coefficients were assumed to be the same across clusters as codon usage in both clusters is still adapting to the same tRNA pool [86].

ROC-SEMPPR predictions of gene expression for each protein-coding sequence were compared to empirical estimates of mRNA abundance using the Spearman rank correlation coefficient. If natural selection on codon usage is strong enough to shape codon usage frequencies, then a statistically-significant positive correlation is expected between predicted and empirical estimates of gene expression [29, 34, 60]. If non-adaptive nucleotide biases vary across the genome, then it is expected that allowing the mutation bias parameters to vary between clusters will significantly improve the correlation between predicted and empirical gene expression. To assess the impact of evolutionary processes that may favor GC-ending codons over AT-ending codons, selection coefficient estimates and mutation bias estimates were modified such that all GC-ending codons were relative to the corresponding AT-ending codon, e.g. GCG was set relative to GCA (as opposed to GCT, the alphabetically last codon for alanine) by subtracting the selection coefficient for GCA from GCG. In this context, a positive selection coefficient (mutation bias) estimate indicates the GC-ending codon is disfavored by natural selection (mutation bias) relative to the AT-ending codon. Similarly, a negative value indicates the GC-ending codon is favored relative to the AT-ending codon.

In addition to ROC-SEMPPR, we also evaluated the impact of variable non-adaptive nucleotide composition biases on the Codon Adaptation Index (CAI), as implemented in CodonW [62]. Briefly, CAI is calculated for each gene as the geometric mean of its synonymous codon weights, which are generally thought to represent how "optimal" or "preferred" a codon is compared to its synonymous. Synonymous codon weights are calculated from the RSCU values calculated for the reference set of genes either known or expected to be high expression [60]. Ribosomal proteins are often used as a reference set, but [61] caution against this. In contrast, CodonW can automatically determine a reference set for calculating CAI using the first principle component of a CA on codon usage, as the first principle component is often correlated with gene expression.

## Examining variation in GC3% along chromosomes

Genes were mapped to their corresponding location on each chromosome indicated within the genome. Genes filtered out prior to analysis with ROC-SEMPPR were also excluded from this analysis. Variation in GC3% content along chromosomes was visualized using a moving average of the per-gene GC3% value across a 20 gene window. Along with this, we calculated the percentage of genes falling into the Higher GC3% cluster within the same 20 gene window. This provides a means to compare GC3% variation along a chromosome with the cluster membership and provides insights into if genes falling into the two clusters determined by the CLARA clustering are also physically clustered within the genome. To deal with non-independence due to autocorrelation, we also calculated the same metrics using non-overlapping 20 gene windows, which were then compared using a Spearman Rank correlation.

## Comparisons across species

To visualize results across 49 species, we used the ggtree R package [87]. The phylogenetic tree was taken from the Supplementary Material of [78]. Phylogenetic independent contrasts (PIC) were used to account for the non-independence when calculating correlations across species [64].

## Supporting information

**S1 Fig. ROC-SEMPPR predictions of gene expression are mostly well-correlated with empirical mRNA abundances across 49 Saccharomycotina yeasts.** Spearman rank correlation coefficients comparing predicted gene expression estimates from ROC-SEMPPR ConstMut fit with empirical estimates of mRNA abundance from RNA-seq data across 49 Saccharomycontina budding yeasts.
(TIF)

**S2 Fig. Comparison of empirical mRNA abundances estimated from disparate RNA-seq measurements.** (A) Distribution of per-gene estimated counts from kallisto. (B-D) Comparison of empirical expression estimates across species using one-to-one orthologs. Correlations represent Spearman rank correlation coefficients.
(TIF)

**S3 Fig. Codon usage in highly-expressed genes is similar across closely-related species.** Across-species comparison of RSCU values calculated from the most highly expressed genes (top 5% based on empirical expression estimates). (A) *Saccharomyces*. (B) *Candida*. (C) *Ogataea*. Correlations represent Spearman rank correlation coefficients.
(TIF)

**S4 Fig. Codon usage differs between high and low expression genes.** Comparison of RSCU values calculated from most highly and lowly expressed genes (top 5% and bottom 5% of expression estimates) for each species. Correlations represent Spearman rank correlation coefficients.
(TIF)

**S5 Fig. Results with heuristic measures of codon usage are similar to those seen with ROC-SEMPPR.** Comparing ability to predict expression using (A) ROC-SEMPPR (ConstMut) and (B) Codon Adaptation Index (CAI). CAI was estimated using CodonW, with the correspondence analysis built-in to CodonW used to identify the reference set.
(TIF)

**S6 Fig. Variation in GC3% along chromosomes of *L. kluyveri*.** Per-gene GC3% content across all *L. kluyveri* chromosomes quantified as a moving average using a 20 gene sliding window (solid line). For each 20 gene window, the percentage of genes assigned to the Higher GC3% regime is also shown (dashed line). Color bars indicate the mutation regimes for Higher and Lower GC3% (blue and right, respectively). The region of high GC3% content on chromosome SAKL0C is the result of an introgression.
(TIF)

**S7 Fig. Impact of using 3 clusters compared to 2 clusters on predicting gene expression data using ROC-SEMPPR.** (A) VarMut_2 model using $k = 2$ clusters. (B) VarMut_3 model using $k = 3$ clusters.
(TIF)

**S8 Fig. Variation in GC3% along chromosomes of *S. uvarum*.** Per-gene GC3% content across all *S. uvarum* chromosomes quantified as a moving average using a 20 gene sliding window (solid line). For each 20 gene window, the percentage of genes assigned to the Higher GC3% regime is also shown (dashed line). Color bars indicate the mutation regimes for Higher and Lower GC3% (blue and right, respectively).
(TIF)

**S9 Fig. Variation in GC3% along chromosomes of *S. cerevisiae*.** Per-gene GC3% content across all *S. cerevisiae* chromosomes quantified as a moving average using a 20 gene sliding window (solid line). For each 20 gene window, the percentage of genes assigned to the Higher GC3% regime is also shown (dashed line). Color bars indicate the mutation regimes for Higher and Lower GC3% (blue and right, respectively).
(TIF)

**S10 Fig. Variation in GC3% along chromosomes of *C. albicans*.** Per-gene GC3% content across all *C. albicans* chromosomes quantified as a moving average using a 20 gene sliding window (solid line). For each 20 gene window, the percentage of genes assigned to the Higher GC3% regime is also shown (dashed line). Color bars indicate the mutation regimes for Higher and Lower GC3% (blue and right, respectively).
(TIF)

**S11 Fig. Variation in GC3% along chromosomes of *C. dubliniensis*.** Per-gene GC3% content across all *C. dubliniensis* chromosomes quantified as a moving average using a 20 gene sliding window (solid line). For each 20 gene window, the percentage of genes assigned to the Higher GC3% regime is also shown (dashed line). Color bars indicate the mutation regimes for Higher and Lower GC3% (blue and right, respectively).
(TIF)

**S12 Fig. Variation in GC3% along chromosomes of *C. parapsilosis*.** Per-gene GC3% content across all *C. parapsilosis* chromosomes quantified as a moving average using a 20 gene sliding window (solid line). For each 20 gene window, the percentage of genes assigned to the Higher GC3% regime is also shown (dashed line). Color bars indicate the mutation regimes for Higher and Lower GC3% (blue and right, respectively).
(TIF)

**S13 Fig. Variation in GC3% along chromosomes of *O. parapolymorpha*.** Per-gene GC3% content across all *O. parapolymorpha* chromosomes quantified as a moving average using a 20 gene sliding window (solid line). For each 20 gene window, the percentage of genes assigned

to the Higher GC3% regime is also shown (dashed line). Color bars indicate the mutation regimes for Higher and Lower GC3% (blue and right, respectively).
(TIF)

**S14 Fig. Variation in GC3% along chromosomes of *O. polymorpha*.** Per-gene GC3% content across all *O. polymorpha* chromosomes quantified as a moving average using a 20 gene sliding window (solid line). For each 20 gene window, the percentage of genes assigned to the Higher GC3% regime is also shown (dashed line). Color bars indicate the mutation regimes for Higher and Lower GC3% (blue and right, respectively).
(TIF)

**S15 Fig. Variation in GC3% along chromosomes of *O. methanolica*.** Per-gene GC3% content across all *O. methanolica* chromosomes quantified as a moving average using a 20 gene sliding window (solid line). For each 20 gene window, the percentage of genes assigned to the Higher GC3% regime is also shown (dashed line). Color bars indicate the mutation regimes for Higher and Lower GC3% (blue and right, respectively).
(TIF)

**S16 Fig. Physical clustering of genes assigned to the same mutational cluster is weaker in species for which VarMut model performed worse.** Comparing the average GC3% and percentage of genes assigned to Higher GC3% cluster using 20-gene non-overlapping windows across all chromosomes for a species. (A) *S. cerevisiae*. (B) *C. parapsilosis*. (C) *O. methanolica*.
(TIF)

**S17 Fig. Comparison of selection coefficients estimated using the ConstMut and VarMut model.** Selection coefficients are modified such that they represent selection of GC-ending codons relative to AT-ending codons, i.e. NNG relative to NNA or NNC relative to NNT. (A) Scatter plot showing the effect of intragenomic mutation bias on misidentifying or overestimating the strength of selection on GC-ending codons relative to AT-ending codons in *C. albicans*. (B) Distribution of log fold changes of selection coefficients between VarMut and ConstMut models.
(TIF)

**S18 Fig. Comparison of mutation biases between the Lower and Higher GC3% clusters estimated using the VarMut model.** Mutation biases are modified such that they represent mutation bias of GC-ending codons relative to AT-ending codons, i.e. NNG relative to NNA or NNC relative to NNT. (A) Scatter plot showing the difference in mutation bias between the two clusters used in the VarMut model for *C. albicans*. (B) Distribution of log fold changes of mutation bias between clusters used in VarMut Model.
(TIF)

**S19 Fig. Variation in GC3% along chromosomes of *L. thermotolerans*.** Per-gene GC3% content across all *L. thermotolerans* chromosomes quantified as a moving average using a 20 gene sliding window (solid line). For each 20 gene window, the percentage of genes assigned to the Higher GC3% regime is also shown (dashed line). Color bars indicate the mutation regimes for Higher and Lower GC3% (blue and right, respectively).
(TIF)

**S20 Fig. Model selection of VarMut and ConstMut models.** Comparison of ΔDIC to (A) differences in prediction of gene expression data between the VarMut and ConstMut models and (B) the differences of the median GC3% values of the Higher GC3% and Lower GC3% clusters.
(TIF)

**S1 Table. List of species and accessions for RNA-seq data used in this study.** All data were download from NCBI's Sequence Read Archive, except for *L. kluyveri* (European Nucleotide Archive).
(TSV)

## Acknowledgments

The authors would like to thank E.W.J. Wallace and M.A. Gilchrist for helpful discussions over the course of this project.

## Author Contributions

**Conceptualization:** Alexander L. Cope, Premal Shah.

**Data curation:** Alexander L. Cope.

**Formal analysis:** Alexander L. Cope.

**Investigation:** Alexander L. Cope.

**Methodology:** Alexander L. Cope, Premal Shah.

**Software:** Alexander L. Cope.

**Supervision:** Premal Shah.

**Visualization:** Alexander L. Cope.

**Writing – original draft:** Alexander L. Cope, Premal Shah.

**Writing – review & editing:** Alexander L. Cope, Premal Shah.

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
