## [Decision Letter · Decision Letter 0]

31 Dec 2021

Dear Dr Cope,

Thank you very much for submitting your Research Article entitled 'Intragenomic variation in mutation biases causes underestimation of selection on synonymous codon usage' to PLOS Genetics.

The manuscript was fully evaluated at the editorial level and by independent peer reviewers. The reviewers appreciated the attention to an important problem, but raised some substantial concerns about the current manuscript. Based on the reviews, we will not be able to accept this version of the manuscript, but we would be willing to review a much-revised version. We cannot, of course, promise publication at that time.

If you decide to revise the manuscript for further consideration at PLOS Genetics, please aim to resubmit within the next 60 days, unless it will take extra time to address the concerns of the reviewers, in which case we would appreciate an expected resubmission date by email to plosgenetics@plos.org.

[LINK]

We are sorry that we cannot be more positive about your manuscript at this stage. Please do not hesitate to contact us if you have any concerns or questions.

Yours sincerely,

Justin C. Fay

Associate Editor

PLOS Genetics

Kirsten Bomblies

Section Editor: Evolution

PLOS Genetics

The reviewers were mostly positive, but also brought up important concerns. The enthusiasm was higher for the identification of intra-genomic variation in bias and potential causes compared to model mis-specification and the resulting differences in estimates of selection. I agree with the reviewers that it is important to know whether the observation can be explained by BGC, introgression or neither of these two. Currently, introgression explains one species but the reviewers question whether it may explain others and whether BGC is an adequate explanation for others. While it may not be possible to be definitive in each case, the reviewers do point out that the plausibility of these explanations could be examined in a number of ways. I also think it important to address the statistical comparison of the two models.

Reviewer's Responses to Questions

**Comments to the Authors:**

Reviewer #1: This manuscript analyzes codon usage bias (CUB) in 49 species of yeast, with a focus on the influence of genome-heterogeneous substitution processes. It is argued that a failure to model this heterogenity can lead to misleading inferences regarding the strength of CUB.

I found some of the analyses and results interesting - particularly, the contrasted pattern between closely related species shown in figure 1. I have a number of remarks regarding the writing and the overall organisation of the manuscript. Importantly, I feel like the main message of the manuscript is not particularly unexpected. Model misspecification is well identified as an important statistical problem in evolutionary genomics in general, heavily discussed in the molecular phylogenetic and population genomic literature. That this problem can also apply to model-based CUB analysis, as demonstrated here, is not surprising. The manuscript uses a specific method, ROC-SEMPER, which although impressively sophisticated, has not been applied to many data sets so far (28 cites since 2015). It is unclear whether the detected biases would also affect, and how much, other approaches to the analysis of CUB. Finally, the reasons for the discrepant patterns among closely-related species are only superficially considered in the manuscript, while I see this as one of the most intriguing aspects of the study. Below I'm providing a couple of more detailed comments, which might help, I hope, improve the manuscript.

- Manuscript organization 1

49 yeast genomes are analysed independently using the ROC SEMPER method. The main analysis only considers 8 of these, out of which 5 are problematic. Then, results from the whole 49 are shown, and we get to know that the remaining 41 are not strongly, or at all, affected by the detected bias. I find this way of presenting the data somewhat misleading in leaving the impression, throughout the reading, that the bias affects a large proportion of the analysed species, whereas in reality this proportion is 10%. I see no good justification for such a progressive introduction of the results.

- Manuscript organization 2

Some additional writing effort is needed if the Methods section is to be located after the Results and Discussion sections, as it currently is. A brief presentation of the data and an explanation of the methods should be inserted at the begining of the Results section, which currently is uneasy to follow - the reader needs to go back and forth between Results and Methods. Generally, I think the ROC SEMPER method and underlying model parameters would deserve to be inroduced in more detail.

- Analysis

An unsupervised classificaton method is used to define two groups of genes per species. Is there any particular reason why two groups were chosen, not 3 or more? Is there a way to assess the significance of the partitioning? The two groups are called low-GC and high-GC, and interpreted throughout as reflecting an inherent within-genome variation in GC-content; so why not simply partition the data based on gene GC3?

- Interpretation 1

The fact that genes of similar GC-content tend to be clustered is interpreted as supporting the hypothesis that GC-biased gene conversion (BGC) explains the among-genes heterogeneity in GC-content. I am not sure I'm following the logic here. BGC is a recombination-associated mechanism, so I would expect BGC to result in such a clustering if recombination was genome-heterogeneous, and correlated to gene GC-content. Do we have data on the recombination map of some of these species? Or reasons to believe that clusters of GC-rich genes fall in high-recombining regions? Also at odds with this explanation is the fact that BGC is a well documented process in S. cerevisiae (eg Mancera et al 2008 Nature; Harrison & Charlesworth 2011 MBE, Lesecque et al 2013 MBE), but S. cerevisiae is identified as a non-problematic, GC-homogeneous species in this study.

- Interpretation 2

In one previously-studied species, Lachancea kluyveri, lateral gene transfer (introgression) was identified as the cause of GC-content heterogeneity. The text says: "For example, both L. kluyveri and L. thermotolerans show a substantial improvement in the predictions of gene expression with the VarMut model; however, only the former contains the introgressed region on one of its chromosomes [48]." This is not a correct statement since L. thermotolerans is not analysed in ref 48. Couldn't GC-content heterogeneity be explained by introgression (of smaller pieces) in this and other species as well? Given the phylogenetic proximity of these species to species with a ~homogeneous genome, this would appear as a plausible explnation to me; I do not see why it is disregarded.

- Interpretation 3

The BGC hypothesis is favoured, and sometimes connected to the (in)ability of various species/clades to engage in sexual reproduction, but I must say I found the whole rationale confusing. A better job could be done of informing the reader of what is known and unknown regarding sex in these 49 species, and whether this relates to the observed patterns.

- Wording 1

The word "isochores" is used in the introduction to describe the existence of GC-content heterogeneity in some vertebrate and yeast species. I would suggest refraining from using this term as it has a long history of adding confusion to the literature - e.g. see https://en.wikipedia.org/wiki/Isochore_(genetics).

- Wording 2

GC-biased gene conversion (BGC) is identified throughout as a mutation bias. This is not correct. BGC is a segregation bias which tends to favour the increase in frequency and eventual fixation of GC alleles in high-recombining regions of the genome. From a population genetic viewpoint, BGC acts similarly to selection, not mutation, although no fitness effect is involved (eg see work by Duret). I understand that the effect of BGC is likely captured by the mutation bias parameters of the ROC SEMPER method, but this is no good reason to call BGC a mutation bias - it is not.

Reviewer #2: The manuscript by Cope and Shah performs a series of analyses of codon usage bias in genomes of budding yeast species to show that variation in mutational biases within each genome can lead to underestimation of selection on synonymous codon usage. The manuscript is generally clearly written and to the point and fits well with the scope of PLoS Genetics. The results of the study are somewhat intuitive and the work significantly contributes to our understanding of codon usage bias and of mutational bias in eukaryotic genomes.

I have one major concern. The authors test two different models, one that assumes constant mutation bias across the genome (ConstMut) and one that assumes variable mutation bias across the genome (VarMut). For 5/8 species tested, the VarMut model gives a better fit between empirical and predicted expression estimates, so the authors conclude that it is better than the ConstMut model. However, the VarMut model has more parameters so how do the authors exclude the possibility that this is due to overfitting? My sense is that there is need to formally compare the two models (e.g., using Akaike Information Criterion or some other formal method).

Minor comments:

- A table that reports the accession numbers of all gene expression datasets of the 49 yeast species used in this study should be provided.

- I spotted quite a few typos (e.g., line 54: "translational", line 118: "suggest", meaning of line 185 not clear, etc.), so some careful checking will be necessary.

Reviewer #3: In the manuscript entitled « Intragenomic variation in mutation biases causes underestimation of selection on synonymous codon usage » Alexander L Cope and Premal Shah demonstrate that biases towards GC can be misinterpreted as positive selection on codon usage in yeasts.

I found the manuscript interesting. My main criticism concerns the writing of the material and methods that did not help me to understand what has been done.

MATERIAL AND METHODS

- How many genes do you have per species ?

- please describe the unsupervised learning method from [52] in one or two extra sentences.

- please define RSCU as you use it at some point (supp fig 2) even if you don’t use it for CA.

- please describe the CLARA algorithm a little bit more in the material and metods (at the moment the description is in the results section)

- please describe in one or two more sentences what ROC SEMPPR does to facilitate the reading. For instance I don’t get the link between the exclusion of CTG codon (serine) and the weak selection hypothesis even if I get that you exclude it because in some species CTG does not translate to serine. Is it really that ROC SEMPPR only assumes weak selection or is it an assumption of codon usage ?

line 349 : you say that you scale the codon usage. Is it GCG/GCA or (GCG -GCA)/GCA ? Please describe precisely tha scaling method.

-please mention somewhere what are the 49 species.

RESULTS

-line 101 : it seems that S. cerevisiae is the outlier species but there is also an other species at 0.42 that could be an outlier. What is the statistical test that lead to the outlier/not-outlier classification ?

- line 116-119 : I don’t see the link between the supp fig 3 and the text : are they the same codons accross species ?

- line 125 : epigenetics could also lead to differential possibility of mutations

-line 140: super interesting !! great to read!

- line 149-151:you directly link expression prediction and selection. Please rephrase to explicit the step taken here.

-figure 2: what does each point represent?

- line 163 : which figure ?

- Figure 3 : I don’t know the proportion of genome that is outside genes. If sufficient, it would be interesting to compare GC3 content to GC content outside genes as gBGC should affect similarly GC3 (under weak selection) and GC outside genes.

- Figure 5A : how can you be less good with more parameters (you can not improve the predictions but how can you decrease the correlation ? I think I miss what is a correlation shift. Again the material and methods must be expanded.

CONCLUSION/DISCUSSION

- Please explicit in further details what it implies with efficiency of gBGC or other mechansisms such as mutational biases and the fistinction between VarMut and ConstMut models per species. Is there gBGC in S.cerevisiae and in the 7 other species when they Var and Const agree/disagree etc.

GENERAL COMMENTS

- I am always confused when authors say that gBGC is a mutation bias. gBGC does not create new mutations but is rather a fixation/substitution bias. When there is a GC in front of a AT, gBGC occurs and tends to favor the spread of the GC polymorphisms up to fixation. To that respect I would rephrase the entire manuscript (it does not affect the global conclusion that gBGC affect the efficacy of positive selection identification).

- I am missing some inter species comparisons. There must be genes highly express in all the species (housekeeping genes) → do they have the same codon usage/ constraints in GC content ? I ask this question because you mention several times that close species have different GC-constraints, I wonder whether it concerns the entire genomes or solely the species-specific highly express genes. I think this would be an interesting analysis to perform if it does not take too much time.

- You barely mention recombination in your paper even if you discuss a lot gBGC. If it is not too difficult, it would be super interesting and would help validating your work to compare in S.cerevisiae the localisation of the high-GC3 clusters to its recombination map (from Mancera for instance but there are more recent work probably). You could check whether high-GC3 clusters are in high recombining region and thus, under strong gBGC.

- I would emphesize more the unsupervised clustering approach, I think this is great and a real plus-value of your work.

I did not check the english spelling as it is not my maternal langage.

**Have all data underlying the figures and results presented in the manuscript been provided?**

Reviewer #1: **No: **

Reviewer #2: **No: **A table that reports the accession numbers of all gene expression datasets of the 49 yeast species used in this study should be provided.

Reviewer #3: Yes

PLOS authors have the option to publish the peer review history of their article (what does this mean?). If published, this will include your full peer review and any attached files.

Reviewer #1: No

Reviewer #2: No

Reviewer #3: No

---

## [Editor Report · Decision Letter 1]

24 Apr 2022

Dear Dr Cope,

Thank you very much for submitting your Research Article entitled 'Intragenomic variation in non-adaptive nucleotide biases causes underestimation of selection on synonymous codon usage' to PLOS Genetics.

The manuscript was fully evaluated at the editorial level and by independent peer reviewers. The reviewers appreciated the attention to an important topic but identified some concerns that we ask you address in a revised manuscript

We therefore ask you to modify the manuscript according to the review recommendations. Your revisions should address the specific points made by each reviewer.

[LINK]

Yours sincerely,

Justin C. Fay

Associate Editor

PLOS Genetics

Kirsten Bomblies

Section Editor: Evolution

PLOS Genetics

Your revised work has addressed the reviewer comments. However, there are a few minor issues that I would like to give you a chance to address before moving forward to accepting the paper.

1. On line 284 you say "we expanded our analysis to 49 species". However, you already mentioned the analysis is of 49 species. It would be better to say you returned to analysis of all 49 species.

2. It is understandable, especially given the Liu (2017) paper, that you don't attempt to attribute the %GC content variation to a specific evolutionary force, such as gBGC. While the complications regarding this issue were clear in the reviewer response, I think it is important to relay these to the reader in the discussion as this is a natural question for a reader. Currently, you state that gBGC is present in Sc and reference Liu in parenthesis, and later mention that recombination rates are not available for most species. I would encourage you to more directly address this issue by stating (e.g. around line 351) that it would be hard/complicated to attribute %GC3 to specific biological processes, with Liu paper raising questions about the impact of gBGC, the weak relationship between GC3 (Harrison 2011) and recombination, and the absence of recombination rates for most species.

---

## [Editor Report · Decision Letter 2]

13 May 2022

Dear Dr Cope,

We are pleased to inform you that your manuscript entitled "Intragenomic variation in non-adaptive nucleotide biases causes underestimation of selection on synonymous codon usage" has been editorially accepted for publication in PLOS Genetics. Congratulations!

Yours sincerely,

Justin C. Fay

Associate Editor

PLOS Genetics

Kirsten Bomblies

Section Editor: Evolution

PLOS Genetics

Comments from the reviewers (if applicable):

**Data Deposition**

http://datadryad.org/submit?journalID=pgenetics&manu=PGENETICS-D-21-01451R2

**Press Queries**

---

## [Editor Report · Acceptance letter]

10 Jun 2022

PGENETICS-D-21-01451R2 

Intragenomic variation in non-adaptive nucleotide biases causes underestimation of selection on synonymous codon usage 

Dear Dr Cope, 

We are pleased to inform you that your manuscript entitled "Intragenomic variation in non-adaptive nucleotide biases causes underestimation of selection on synonymous codon usage" has been formally accepted for publication in PLOS Genetics! Your manuscript is now with our production department and you will be notified of the publication date in due course.

With kind regards,

Zsofi Zombor

PLOS Genetics

On behalf of:
